# Human Papillomavirus Infection in Partners of Women Attending Cervical Cancer Screening: A Pilot Study on Prevalence, Distribution, and Potential Use of Vaccines

**DOI:** 10.3390/vaccines13020172

**Published:** 2025-02-11

**Authors:** Arianna Sucato, Nicola Serra, Michela Buttà, Leonardo Di Gregorio, Daniela Pistoia, Giuseppina Capra

**Affiliations:** 1Department of Health Promotion, Mother and Child Care, Internal Medicine, Medical Specialties G. D’Alessandro, University of Palermo, 90127 Palermo, Italy; arianna.sucato@unipa.it (A.S.); michela.butta@unipa.it (M.B.); 2Department of Neuroscience, Reproductive Sciences and Dentistry—Audiology Section, University of Naples Federico II, 80131 Naples, Italy; nicola.serra@unina.it; 3Unit of Urology, Polyclinic Hospital, Via del Vespro 133, 90127 Palermo, Italy; leondigregorio@gmail.com; 4Microbiology and Virology Unit, University Hospital Policlinic, P. Giaccone, 90127 Palermo, Italy; daniela.pistoia@policlinico.pa.it

**Keywords:** male, HPV, couples, cervical cancer screening, HPV vaccine, human papillomavirus, HPV infection

## Abstract

**Background/Objectives**: Human Papillomavirus (HPV) cross-infection among couple’s partners is a widespread event and could lead to persistent infections. Unfortunately, the influence of male sexual partners of HPV-positive women on their cervical lesions and the potential role of HPV vaccines have been under-investigated. We evaluated the HPV prevalence in male partners of HPV-infected women, focusing on the possible correlation between partners’ cervical lesions and the role of HPV vaccination. **Methods**: Two samples, genital and urethral swabs, were collected for each of the 90 patients recruited. HPV-DNA detection was carried out by the Allplex HPV28 detection assay. **Results**: HPV prevalence was 66.7% (60/90); high-risk HPV (hrHPV) genotypes were detected in 90% (54/60) cases and multiple infections in 55% (33/60). The most frequent hrHPVs were HPV31 (*p* = 0.0265) and HPV52 (*p* = 0.002), found in 18.3% (11/60) of cases, and HPV53 (*p* = 0.0116) in 16.7% (10/60). Statistical analysis showed a higher probability of a less severe cytological diagnosis with the increase in the number of genotypes detected (*p* = 0.0146). Among the HPV-positive partners of females with cervical lesions, 18.7% (6/32) and 62.5% (20/32) had vaccine genotypes of the quadrivalent and nonavalent vaccines, respectively. The nonavalent vaccine showed a significantly higher potential to prevent lesions (62.5% vs. 18.7%, *p* = 0.0001), with an absolute additional potential impact (AAI) of 31.1% in histological and 32.4% in cytological diagnoses. **Conclusions**: These preliminary results provide new insight into the correlation between the number of partner genotypes and the severity of cervical lesions and show promising results for the preventive potential of vaccinating male partners.

## 1. Introduction

Human papillomavirus (HPV) is one of the most widespread sexually transmitted infections (STIs) worldwide, with similar global prevalence rates in men and women, ranging from 3.5–45% and 2–44%, respectively [1]. Almost 80% of sexually active men and women can become infected at least once in their lifetime, as infection can occur not only through sexual intercourse but also through rubbing or touching the skin surfaces and mucous membranes [1,2,3,4,5]. In this framework, the likelihood of contracting a sexually transmitted infection such as HPV has increased owing to a lower age of first sexual intercourse and a higher number of sexual partners recorded in recent decades [6].

Papillomaviruses of human concern are included into five genera (α, β, γ, μ, and ν) grouped on the sequence similarity of the highly conserved L1 gene [7,8,9].

To date, more than 200 different HPV genotypes have been described, classified as high-risk (hrHPV) based on their association with the onset of cervical, vaginal, vulvar, anal, head and neck, and penile cancers, and low-risk (lrHPV) associated with genital warts or condyloma acuminata [10,11]. More often, infections are completely cleared by the immune system over 12–24 months with no clinical complications. However, if the immune system fails to clear the virus, a persistent infection is established, laying the foundation for the development of neoplastic hyperproliferative lesions [12].

The HPV carcinogenic nature, as reported in the literature, is linked to its double-stranded circular genome that encodes the two oncoproteins, E6 and E7, which cooperate to overcome cellular apoptotic pathways by interacting with p53 and pRb tumor suppressor proteins, respectively [10,13,14,15].

In particular, persistent hrHPV infection was associated with the development of squamous intraepithelial lesions (SILs), but also penile intraepithelial neoplasia (PeiN), which are precursors to cervical cancer in women and penile cancer in men [16,17]. In addition, lifestyle factors such as long-term use of oral contraceptives, smoking, compromised immune status, or other concurrent sexually transmitted infections may favor neoplastic proliferation [2].

The majority of previous studies have focused on HPV infection in women because of its role in the development of over 90% of cervical cancers [2,16]. Against this background, the literature on the management of women with an intraepithelial lesion is extensive, while studies on their male sexual partners are scarce [17]. Indeed, mortality and morbidity due to HPV infection in men are low [18]. Nonetheless, in recent years, there has been a growing interest in improving the knowledge about the weight of HPV infection and related diseases in men, given its implication with genital warts, penile, anorectal, and oropharyngeal cancers, and especially for the role played in the transmission to their female sexual partners [16,18]. In addition, transmission can take place easily between sexual partners, and in many cases, a couple can have several undetected transmission events [18,19,20,21].

On the other hand, it has been widely suggested that men may act as reservoirs or vectors of HPV infection since infection in these patients is often subclinical or latent, so they do not show clinical lesions [22]. This absence of symptoms and the lack of screening programs for men lead them to be tested only if their partner is positive or to check for couple fertility problems, increasing the transmission rate [18,23,24,25,26].

An effective weapon to prevent HPV infection and related consequences is vaccination. Up to today, the Food and Drug Administration (FDA) and the European Medicines Agency (EMA) approved three HPV vaccines: the bivalent Cervarix, which covers against infection with HPV16 and 18; the quadrivalent Gardasil, which protects against HPV6, 11, 16, and 18, and the nonavalent Gardasil 9, against HPV6, 11, 16, 18, 31, 33, 45, 52, 58 [27,28,29].

In this context, males typically showed poor adherence to vaccination despite it being a valid way to prevent HPV-related malignancies. For this reason, strategies such as HPV vaccination campaigns, which generally target women, should be implemented for men to protect not only themselves but also their sexual partners [18,30,31,32].

Approximately 50–77% of regular male partners of women who are HPV-positive or with cervical lesions, or both, have subclinical HPV infection, which may be an indirect causal factor for cervical neoplasia [30,33]. To the best of our knowledge, the asymptomatic nature of male HPV infection and men’s sexual behavior could influence the chance of transmission to female partners and, thus, the risk of developing cervical lesions [34,35]. A risk that is also increased by the cross-infection between the couple’s partners, as it is more likely than other cases to result in persistent HPV infection [18,25].

Considering the above, in this prospective observational study, we evaluate the prevalence of genital HPV infection and genotype distribution among a cohort of 90 male partners of HPV-infected women. Specifically, we aimed to shed light on the controversial and under-researched impact of male HPV infection on the female partner’s cervical health by evaluating the correlation between male HPV positivity and female cervical lesions, with a focus on the potential influence of HPV vaccines.

## 2. Materials and Methods

### 2.1. Study Population and Samples Collection

The study population consisted of 90 male patients aged 23 years or older who presented for HPV testing at the Microbiology and Virology Laboratory of the Polyclinic “P. Giaccone” Hospital, Palermo, Department of Health Promotion, Mother and Child Care, Internal Medicine and Medical Specialties (PROMISE). The reason for testing was that their female partners had tested positive for hrHPV in the Cervical Cancer Screening program, and male patients were included following a continuous criterion.

Two samples were performed, genital and urethral swabs, collected by the urologists of the Urology of Polyclinic “P. Giaccone” Hospital following a urological examination.

The genital swab collected cells from the dorsal/ventral surface of the penile shaft and cells from the inner foreskin, coronal sulcus, frenulum, and glans, sampled with a saline-prewetted cytobrush placed in a tube containing 3 mL of phosphate-buffered saline (PBS). Five to six swab/cytobrush movements were performed on each penile site.

The urethral swab (Puritan HydraFlock^®^, Puritan medical products, Guilford, CT, USA) collected cells by inserting a thin, saline-soaked brush into the urethra, rotating it 360 degrees, removing, and discharging the brush into a separate vial with 3 mL of PBS.

Male patients were advised to wash their genitals with water only without using detergents on the day of the examination.

All the patients brought with them the female partner’s cytological and histological diagnosis. The cytology was reported according to the Bethesda System 2014 [36].

In particular, cytology plays the role of a “screening test” in the Cervical Cancer Screening Program, whereas histology is a diagnostic test. Indeed, the program offers a first step, which includes HPV-DNA test and cytology. In the case of positivity to both, the patient will have access to colposcopy and, if necessary, to histology examination, according to the Cervical Cancer Screening guidelines of Italy. On the other hand, in the case of only HPV-DNA test positivity, the patient will be re-examined after 12 months. If the repeated HPV-DNA test gives a positive result, the patient will promptly be sent for colposcopy and histology exams.

### 2.2. Ethical Statement

All study participants expressed their agreement to participate in socio-behavioral interviews. The study was conducted in accordance with the Helsinki Declaration. The informed consent and the study protocol were approved by the Institutional Review Board at the Polyclinic, University of Palermo, Italy (Approval number #20/2024 of 30 July 2024). No financial compensation was provided for the participants.

### 2.3. Samples Processing and DNA Extraction

Each sample was subjected to centrifugation at 1600 rpm for 10 min, and the pellet was resuspended in 1 mL of phosphate-buffered saline, PBS (EuroClone S.p.A., Milan, Italy) to isolate the cellular component. A second centrifuge was performed at 13,000 rpm for 5 min to discard the supernatant and keep the pellet by storing it at −20 °C or processing immediately. The pellets were resuspended in a volume of 200 μL PBS and used for DNA extraction using the magnetic bead technology of ELITe InGenius^®^ SP200 extraction reagent by the ELITe InGenius^®^ automated extractor (Elitechgroup, Turin, Italy).

### 2.4. HPV Detection and Genotyping

HPV-DNA evaluation was carried out using the Allplex HPV28 detection assay (Seegene Inc., Seoul, Republic of Korea), which identifies 20 genotypes classified as hrHPV (HPV16, 18, 26, 31, 33, 35, 39, 45, 51, 52, 53, 56, 58, 59, 66, 68, 69, 70, 73 and 82) and eight genotypes classified as lrHPV (HPV6, 11, 40, 42, 43, 44, 54 and 61) basing on the classification of the International Agency for Research on Cancer (IARC). Samples testing negative for HPV were subjected to two-step nested PCR, with the PGMY09/11 primer pair followed by GP05+/GP06+ primers, to evaluate the true negative nature of the samples, as described elsewhere [37]. For samples testing HPV positive from PCR, genotypes were defined with INNOLiPA^®^ HPV Genotyping Extra II (Fujirebio, Tokyo, Japan), which identifies 32 genotypes, divided into 23 classified as hrHPV (HPV16, 18, 31, 33, 35, 39, 45, 51, 52, 56, 58, 59, 67, 68, 26, 53, 66, 70, 73, 82) and nine classified as lrHPV (HPV6, 11, 40, 42, 43, 44, 54, 61, 62, 81, 83, 89).

### 2.5. Statistical Analysis

Data are shown as numbers and percentages for categorical variables, while continuous data are expressed as mean and standard deviation (SD) or median and interquartile range (IRQ = [Q1, Q3]).

To compare two mutually exclusive proportions or percentages in groups, the binomial test (B) was used. The chi-square goodness of fit was used to evaluate significant differences among three or more modalities of a variable.

The evaluation of significant differences in proportions or percentages between the two groups was performed with the chi-square test (C) and Fisher’s exact test (F). The multiple comparison chi-square and Fisher’s exact test were used to define significant differences among three or more figures for unpaired data. If chi-square or Fisher’s exact test results were significant (*p*-value < 0.05), the post hoc test was performed using the Adjusted Standardized Residuals and the Z-test (Z). When the chi-square test was not appropriate, it was substituted with Fisher’s exact test. The McNemar test was used to analyze the difference between paired proportions, particularly to test the impact of quadrivalent rather than nonavalent vaccines in females with cervical lesions.

The Shapiro–Wilk test was performed for normal distribution. The differences between the two means of unpaired data were analyzed using the *t*-test (T). Before performing the *t*-test, if the *p*-value was low (*p* < 0.05), the variances of the two samples cannot be assumed to be equal; in this case, the Welch test for correction of unequal variances was used. Alternatively, if the distributions were not normal, the Mann–Whitney test (MW) was performed. Specifically, mean ranks were described when tests on medians showed a significant difference, and the medians were equal. If the variable distributions were not Normal, the relationship between the two parameters was calculated using the Spearman correlation coefficient rho. For this step, we define the variable “diagnosis (Histological/cytological)” and considered the following scale according to disease degree: negative = 0, ASC-US = 1, LSIL = 2, ASC-H = 3, HSIL = 4, Adenocarcinoma = 5.

Additionally, since some statistical tests include small data, we performed the power analysis for each statistical test using the effect size. Particularly, the effect size was computed by phi coefficient for categorical variables, by *η*^2^ and *r* for the non-parametric test (Mann–Whitney test and Wilcoxon signed-rank test, respectively), and by Cohen’s *d* index (paired and unpaired *t*-test).

The absolute additional potential impact (AAI) of the nonavalent vaccine, i.e., the proportion of additional cases potentially prevented by the nonavalent vaccine compared to the quadrivalent vaccine, was calculated as [(*n*_nonavalent_ − *n*_quadrivalent_)/*N*] × 100, with *n* being the number of lesions/infections potentially prevented and *N* the total number of lesions/infections [38].

The statistical analysis was performed using the Matrix Laboratory (MATLAB) analytical toolbox version 2008 (MathWorks, Natick, MA, USA) for Windows at 32 bits.

Finally, all tests with *p*-value (*p*) < 0.05 were considered significant.

## 3. Results

The study cohort consisted of 90 male partners of women infected with high-risk human papillomavirus (hrHPV). Their ages ranged from 23 to 76 years, with a mean age of 43.0 years (standard deviation of 11.8 years). The sociodemographic and sexual history variables, comparing the negative and positive HPV subgroups, are shown in Table 1.

Only 5.6% (5/90) of participants (all HPV-positive) exhibited symptoms (condylomas) during the examination. 48.9% (44/90) and 71.1% (64/90) reported smoking and drinking alcohol, respectively. Among smokers, 10% (9/90) smoked less than 10 cigarettes per day, while 38.9% (35/90) smoked more than 10 cigarettes per day. Among drinkers, 51.1% (46/90) drank alcohol rarely, and 20% (18/90) drank frequently.

On average, people reported having had about 19 sexual partners in their lifetime, with a mean age at first sexual intercourse of about 17 years old. At the examination time, couples had been together for a mean of 10 years, with 80% (72/90) reporting monogamy since the start of their relationship, and 40% (36/90) of men reported using condoms.

Regarding education, 26.7% (24/90) had completed lower secondary school, 48.9% (44/90) upper secondary school, and 24.4% (22/90) tertiary education.

Moreover, no one reported having been vaccinated against HPV.

The statistical analysis found that there were no significant differences in sociodemographic and sexual history variables between men with and without HPV infection, with one exception. Men with HPV infection were found to have a significantly lower age at first sexual intercourse compared to those without HPV infection (median: 16 vs. 17.5, *p* = 0.0076) (Table 1). This significant result was acceptable; in fact, the effect size showed a medium effect (η^2^ = 0.08), i.e., a medium-intensity relationship between the two variables (Table 1).

HPV-DNA was detected in 66.7% (60/90) of genital samples. In particular, hrHPV genotypes, alone or with low-risk genotypes (lrHPVs), were detected in 90% (54/60) cases, whereas lrHPV was only in 10% (6/60). Single and multiple infections were 45% (27/60) and 55% (33/60), respectively (Table 1).

The identified genotypes and their percentages are shown in Figure 1, where we also reported the *p*-values of significantly more frequent genotypes among hrHPV and lrHPV.

The most frequent and statistically significant oncogenic genotypes are HPV31 and HPV52 (*p* = 0.002), found in a percentage of 18.3% (11/60) and HPV53 (*p* = 0.0116) in 16.7% (10/60). Frequent but not significant HPV51 and HPV66 in 13.3% (8/60) and HPV16 in 11.7% (7/60) of cases. Instead, the non-oncogenic genotype most frequently and statistically significant detected was HPV42 (*p* < 0.0001) in 21.7% (13/60) cases, followed by HPV61 frequent but not significant in 10% (6/60) (Figure 1).

Of the 90 patients, 5.6% (5/90) were symptomatic, and 80% (4/5) of those patients had multiple infections with hrHPVs and lrHPVs. The most common lrHPV types found in these low-grade condylomatous lesions were HPV42, HPV61, and HPV6.

We also analyzed the association between men’s HPV infection and the clinical lesions of their female sexual partners. Of the 90 hrHPV-positive female sexual partners of the male patients included in this study, a significant proportion (75.6%, 68/90) exhibited abnormal cervical cytology (Table 2).

Specifically, 8.9% (8/90) had atypical squamous cells of undetermined significance (ASC-US), 35.6% (32/90) a low-grade squamous intraepithelial lesion (LSIL), 20% (18/90) atypical squamous cells—cannot exclude high-grade squamous intraepithelial lesion (ASC-H), and 11.1% (10/90) a high-grade squamous intraepithelial lesion (HSIL). Conversely, 24.4% (22/90) of the female partners had normal cytology. In addition, among cytological diagnoses, we found a significant presence of LSIL in the total sample (35.6%, *p* < 0.0001) and in the partners of the HPV-positive males’ subgroup (40%, *p* < 0.0001). In contrast, no significant differences were observed in the partners of HPV-negative males’ subgroup (*p* = 0.14).

Thus, 48.9% (44/90) of the abnormal cervical cytological diagnoses were histologically confirmed or had a more severe diagnosis. In particular, of these, 25% (2/8) ASC-US, 83.4% (15/18) ASC-H, and 25% (8/32) LSIL had a more severe histological diagnosis of HSIL, and 25% (8/32) of LSIL and 100% (10/10) of HSIL were histologically confirmed. Among the total histological diagnoses, LSIL and HSIL were identified in 10% (9/90) and 38.9% (35/90), respectively (Table 2).

For the sake of clarity, histological lesions were identified in 54.4% (32/60) of the partners of the HPV-positive males and 43.4% (13/30) of the partners of the HPV-negative males.

No association was found between either histological or cytological diagnosis and HPV-positive male partners. An interesting finding was that one of the 22 patients with negative cytology but positive HPV testing for more than 12 months was diagnosed with adenocarcinoma in situ on histological examination.

About histological diagnoses, we found a significant presence of HSIL for the total sample (38.9%, *p* < 0.0001) and in both the subgroups of partners of HPV-positive (40%, *p* < 0.0001) and HPV-negative males (36.7%, *p* < 0.0001).

From power analysis, we observed that all the significant statistical tests showed a large effect size, confirming that the results of the statistical tests had a low probability of being influenced by statistical biases.

In Table 3, we performed two statistical analyses. The first analysis described in the last column was performed among columns, and the second was performed for each column (analyses into groups). For both analyses, we considered only lesions (ASC-US, LSIL, ASC-H, HSIL, and adenocarcinoma).

Table 3 shows a significant correlation between cytological diagnosis and number of genotypes (*p* = 0.0496). ASC-US was more frequent in patients with five and four genotypes (60%, *p* = 0.0003, and 50% *p* = 0.034, respectively), LSIL was more common in patients with two genotypes (63.6%, *p* = 0.0206), and ASC-H was more prevalent in patients with only one genotype (29.6%, *p* < 0.0320). Focusing on patients with two genotypes, LSIL was the most frequent diagnosis (*p* < 0.0001).

Regarding histological diagnosis, HSIL diagnosis was more frequent in patients with one genotype (70%, *p* < 0.0001). No significant association between the number of genotypes and the other histological diagnoses was found (*p* = 0.33).

From power analysis, we observed that all significant statistical tests showed a large effect size, confirming that the statistical tests had a low probability of being influenced by statistical biases.

Figure 2 shows an additional analysis of the possible correlation between diagnosis and number of genotypes performed using Spearman’s correlation coefficient (rho). Considering only lesions, there was a significant correlation between cytological diagnosis and the number of genotypes (rho = −0.36, *p* = 0.0146), but no significant association between histological diagnosis and the number of genotypes was found (rho = −0.05, *p* = 0.79).

Particularly, rho = −0.36 represented a significant negative medium correlation between cytological diagnosis and the number of genotypes.

In Table 4, male single-genotype infections were analyzed with respect to the number of cytological and histological female partner diagnoses, describing their presence or absence in the quadrivalent and nonavalent vaccines.

In particular, among the hrHPV not included in the vaccines, HPV51 appears to be related to 1 case of HSIL and 1 case of LSIL in the respective female partners, HPV56 with 2 cases of HSIL, HPV59, and HPV68 with 1 case of HSIL, and HPV66 and HPV73 with 1 case of LSIL, respectively.

Therefore, concerning the total histological diagnoses of HSIL, 20.8% (5/24) were associated with one genotype not included in the vaccines, whereas the total histological diagnoses of LSIL, 42.8% (3/7) were associated with one genotype not included in the vaccines.

Table 5 shows confirmed histological diagnosis in women associated with the male partner’s genotypes included or not in the vaccines.

Specifically, among the 53.4% (32/60) HPV-positive males with a partner with a histologically confirmed cervical lesion, 18.7% (6/32) harbored at least one of the quadrivalent vaccine genotypes, and 62.5% (20/32) at harbored least 1 of the 9 genotypes of the nonavalent vaccine. These results indicate a significantly higher potential for the nonavalent vaccine to prevent cervical lesions concerning the quadrivalent vaccine (62.5% vs. 18.7%, *p* = 0.0001).

Thus, we computed the Additional Absolute Impact (AAI) to assess the potential impact of the nonavalent vaccine compared to the quadrivalent vaccine. Our analysis revealed that the nonavalent vaccine could prevent an additional 31.1% of cases diagnosed histologically and 32.4% of cases diagnosed cytologically (Table 6). No significant difference was observed (AAI%: 31.1% vs. 32.4%, *p* = 0.89).

Finally, in Table 7, we investigated the possible coverage of vaccination for each disease, considering both cytological and histological diagnoses. For this step, we consider the total sample and two subgroups for cytological and histological diagnosis, represented by negative and positive, i.e., no lesions and lesions (ASC-US, LSIL, ASC-H, HSIL, and adenocarcinoma).

Table 7 highlights no association between a potential quadrivalent coverage and cytological diagnoses. Particularly, the rate of negative/positive of the genotypes not included in the quadrivalent vaccine was statistically equal to the rate of negative/positive of the included genotypes (*p* = 1.0). Similarly, for cytological diagnoses and the nonavalent vaccine (*p* = 0.44). Regarding histological diagnoses, no correlation between quadrivalent (*p* = 0.59) and nonavalent potential coverage (*p* = 0.45) was observed.

Finally, the comparison between positive and negative diagnoses shows a significant presence of positive than negative diagnoses for both histological and cytological diagnoses. Specifically, the significance of histological and cytological diagnoses of the female partners was found in patients with genotypes included and not included in the nonavalent vaccine and in patients with genotypes included in the quadrivalent vaccine. No significance was found in patients with genotypes not included in the quadrivalent vaccine. Finally, from power analysis, we observed that all significant statistical tests showed a large effect size except for the nonavalent not included in cytological diagnosis, where we found a medium effect. These results confirm that the statistical tests had a low probability of being influenced by statistical biases.

## 4. Discussion

The decline in the average age of first sexual intercourse and the increase in the number of lifetime partners has made HPV infection a widespread event, especially between sexual partners [6,18]. In the past years, the influence of the sexual partners of HPV-positive women with cervical lesions has been an under-researched topic but widely debated because of its controversial nature [39]. As has been described in the literature, HPV infection can persist in women after treatment of lesions, prolonging viral transmission to their sexual partners for a long time [40,41,42]. Conversely, the male partner HPV-infected could act as a reservoir of the virus and reinfect his female partner, continuing the promotion of viral transmission [42]. In addition, the issue regarding male infection is exacerbated by the lack of male-specific screening programs, such as cervical cancer screening. Based on this evidence, our study, which analyzed a cohort of 90 male partners of hrHPV-positive women, aimed to provide an upgraded insight into male sexual behavior and its potential association with the risk of their female partners developing cervical lesions, focusing on the possible use of HPV vaccines in preventing these lesions.

Analysis of the sociodemographic and sexual history variables of the study cohort showed a significant difference in the age at first sexual intercourse among HPV-positive and negative patients (*p* = 0.0076) but no significant differences with the other variables.

Our results are consistent with those of Rombaldi et al. [43] for sociodemographic variables. However, they found a significant correlation for the number of sexual partners, while we identified it for age at first sexual intercourse. Unexpectedly, smoking status, a factor known to influence HPV persistence by weakening the immune system and accelerating cell turnover [44,45], did not appear to be significantly associated with HPV positivity in our cohort. The analyzed cohort of patients showed an HPV-positivity rate of 66.7%, but only 5.6% showed symptoms. Asymptomatic infection in men is common, and these data confirm that there is a different pattern of HPV infection and disease manifestation between men and women [17,39].

Our study population revealed HPV31, HPV52, HPV53, and HPV42 as the most prevalent and significant genotypes. In addition, the other two genotypes frequently found in our cohort are HPV51 and HPV66.

Of these genotypes, only HPV31 and HPV52 are included in current vaccine formulations. HPV51, HPV53, and HPV42 have also been identified in other studies, but they are mainly found in women [46,47,48].

Research conducted by Buttà et al. [31] found HPV42, HPV51, HPV53, and HPV66 also in the male population, paying attention to the classification of the International Agency for Research on Cancer (IARC) as “carcinogenic” for HPV51 and “possibly carcinogenic to humans” for HPV53 and HPV66 [11].

Due to the characteristics of cervical cancer screening, whose laboratories use tests that only identify hrHPV without specifying the genotype, we cannot definitively determine specific type concordance between partners.

Despite this, the presence of HPV51 among partners of HPV-positive women was already shown by Bosco et al. [30]. On the contrary, other studies of male partners of HPV-positive women found HPV16 to be the most common genotype [18,39,49]. Given the limited and outdated literature on this type of population, although we also found a good percentage of HPV16 in our patients, we can hypothesize a change in the circulating genotypes following the vaccination of female cohorts. This finding aligns with the phenomenon of type replacement, as described by Chen et al. [50], where vaccination-induced declines in vaccine-type HPV infections lead to increased prevalence of non-vaccine types. While HPV16 remains a serious concern in both sexes, the transmission of other high-risk HPV genotypes from men to women poses a significant risk of female neoplasia [51]. In light of this, investigating the possible association between male HPV genotypes and cervical lesions of their female sexual partners, we found a significant correlation between the number of genotypes involved in male infection and the severity of cervical lesions.

Our results suggest that cytological diagnoses in female partners of men with multiple HPV infections appear to be less severe than those in female partners of men with single infections (*p* = 0.0146). Additionally, we observed a significant association between a frequent histological diagnosis of HSIL in the female partners and infections with only one genotype in the male (*p* < 0.0001). To the best of our knowledge, this is the first study to explore this possible correlation.

These findings could be explained by considering the specific risk groups of male infections. In particular, most multiple-genotype infections in our cohort involve a combination of hrHPVs and lrHPVs. Recent studies [50,52] suggest that women with co-infection of lrHPV and hrHPV have a lower risk of cervical disease. This finding supports the hypothesis of a possible competition mechanism between lrHPVs and hrHPVs in infection, as lrHPVs may act antagonistically by interfering with hrHPV-associated progression in cervical cancer [50,52].

Therefore, based on the evidence available in the literature and our results, we can hypothesize that in our couples, cross-infection phenomena due to the transmission of both hrHPV and lrHPV genotypes could predispose to such a mechanism in the female partner [18,50,52].

However, as previous research has primarily focused on mixed infections in women, and considering the preliminary nature of our data, further investigation is necessary to confirm this hypothesis.

Prophylactic vaccination is acknowledged by the CDC (Centers for Disease Control and Prevention) as the most effective strategy to prevent HPV infection and related manifestations [53]. Indeed, cervical cancer screening offers vaccination after participating in the program because adjuvant vaccination could reduce the risk of cervical lesions [54]. In our analysis, in the subgroup of men positive for the vaccine genotypes with a partner with cervical lesions, the nonavalent vaccine showed greater potential protection against the development of histological diagnoses than the quadrivalent vaccine (*p* = 0.0001). Therefore, through AAI analysis, we estimated the number of additional cases potentially prevented by the nonavalent vaccine compared to the quadrivalent vaccine for histological diagnoses (31.1%) and cytological diagnoses (32.4%).

As far as we know, this is the first study to investigate the impact that vaccinating male partners could have in preventing HPV transmission and subsequent infection in female partners. Until now, previous studies mainly focused on the effectiveness of the nonavalent vaccine in preventing cervical lesions in women, indicating that the transition from quadrivalent to nonavalent vaccine would benefit the prevention of LSIL and HSIL in approximately 90% of cases [28,38,55].

The analysis of genotype distribution and the implementation of vaccination of male sexual partners of HPV-positive women could have a beneficial effect on cervical lesions.

For this reason, increased knowledge of these issues would benefit public health systems by reducing the costs of managing female and male patients, considering that in Italy, HPV-related diseases carried by men were more relevant in terms of economic burden on the total costs [56]. Moreover, identifying advisable genotypes to include in future vaccine formulations could optimize their effectiveness against prevalent HPV genotypes. Data collected in this pilot study provided a first insight into the prevalence of HPV genotypes in an understudied cohort of patients. Overall, to better understand the potential implications of our findings, it seems imperative to expand the patient cohort and identify the genotypes involved in female partner infection to assess concordance within the couple.

## Figures and Tables

**Figure 1 vaccines-13-00172-f001:**
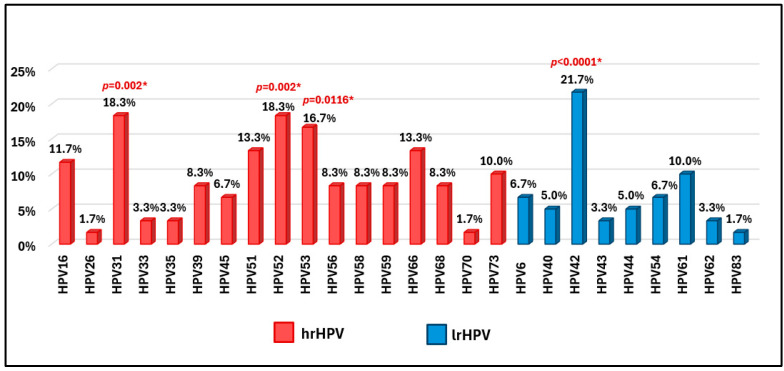
Percentages of hrHPV and lrHPV and most frequent significant genotypes detected in male partners (* = significant test; *p* = *p*-value).

**Figure 2 vaccines-13-00172-f002:**
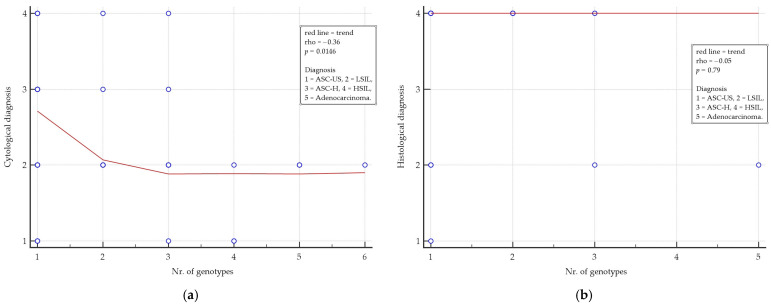
Correlation between female partners’ diagnosis and the number of genotypes detected in males. The red line is the trend line. (**a**) Cytological diagnosis/number of genotypes; (**b**) Histological diagnosis/number of genotypes. The diagnosis scale is described in the Statistical Analysis.

**Table 1 vaccines-13-00172-t001:** General characteristics and lifestyle habits of the study cohort (90 males).

Parameters	Total	Negative to HPV (HPV−)	Positive to HPV (HPV+)	HPV− vs. HPV+	
Study cohort	90	33.3% (30)	66.7% (60)	*p*-value (test)	Effect size
Age Mean (SD) Median (IRQ)	43.0 (11.8) 40.0 (34.0, 51.0)	44.2 (10.3) 44.0 (35.0, 51.0)	42.4 (12.5) 40.0 (33.5, 51.0)	0.36 (MW)	η^2^ = 0.009 trivial effect
Symptoms No Condylomas	94.4% (85/90) 5.6% (5/90)	100% (30) 0.0% (0)	91.7% (55) 8.3% (5)	0.16 (F)	phi = 0.28 medium effect
Smoke No EX <10/DIE ≥10/DIE	42.2% (38) 8.9% (8) 10.0% (9) 38.9% (35)	40.0% (12) 10.0% (3) 6.7% (2) 43.3% (13)	43.3% (26) 8.3% (5) 11.7% (7) 36.7% (22)	0.87 (F)	phi = 0.11 low effect
Alcohol No Rarely Frequently	28.9% (26) 51.1% (46) 20.0% (18)	30.0% (9) 56.7% (17) 13.3% (4)	28.3% (17) 48.3% (29) 23.4% (14)	0.52 (C)	phi = 0.17 low effect
Nr. partners Mean (SD) Median (IRQ)	18.7 (23.3) 10.0 (6.0, 20.0)	18.6 (28.3) 10.0 (5.0, 15.0)	18.8 (20.6) 10.5 (7.5, 22.0)	0.22(MW)	η^2^ = 0.017 low effect
Age at first sexual intercourse Mean (SD) Median (IRQ)	16.8 (2.8) 17.0 (15.0, 18.0)	18.0 (3.5) 17.5 (16.0, 19.0)	16.2 (2.2) 16.0 (15.0, 18.0)	0.0076 * (MW)	η^2^ = 0.080 medium effect
Years with a steady partner Mean (SD) Median (IRQ)	9.7 (10.2) 5.5 (2.0, 14.0)	9.9 (8.9) 7.0 (2.8, 13.3)	10.0 (10.9) 4.0 (2.0, 15.0)	0.65 (MW)	η^2^ = 0.002 trivial effect
Exclusive relationship YES NO	80.0% (72) 20.0% (18)	86.7% (26) 13.3% (4)	76.7% (46) 23.3% (14)	0.27 (C)	phi = 0.16 low effect
Condom use YES NO	40.0% (36) 60.0% (54)	43.3% (13) 56.7% (17)	38.3% (23) 61.7% (37)	0.65 (C)	phi = 0.03 trivial effect
Education level Lower secondary school High school University	26.7% (24) 48.9% (44) 24.4% (22)	20.0% (6) 46.7% (14) 33.3% (10)	30.0% (18) 50.0% (30) 20.0% (12)	0.32 (C)	phi = 0.29 medium effect
Vaccination YES NO	0.0% (0) 100% (90)	0.0% (0) 100% (30)	0.0% (0) 100% (60)	1.0(F)	phi = 1.29 large effect
Risk group hr/lrHPV lrHPV	60.0% (54) 6.7% (6)	─	90.0% (54) 10.0% (6)	─	
Number of genotypes One >1	30.0% (27) 36.7% (33)	─	45.0% (27) 55.0% (33)	─	

* = significant test (*p* < 0.05); MW = Mann–Whitney test; test for Normal distribution was performed using the Shapiro–Wilk test; C = chi-square test; F = Fisher’s exact test; Data were described by mean and SD and median (IQR) or percentage.

**Table 2 vaccines-13-00172-t002:** Cytological and histological diagnoses of the HPV-positive female partners.

Diagnosis	Total	Male Partner HPV-Negative (HPV−)	Male Partner HPV-Positive (HPV+)	HPV− vs. HPV+	
				***p*-Value (Test)**	**Effect Size**
Female partners	90	30	60		
Cytological diagnosis Negative ASC-US LSIL ASC-H HSIL	24.4% (22) 8.9% (8) 35.6% (32) 20.0% (18) 11.1% (10)	30.0% (9) 6.7% (2) 26.7% (8) 26.7% (8) 10.0% (3)	21.7% (13) 10.0% (6) 40.0% (24) 16.7% (10) 11.7% (7)	0.60 (F)	phi = 0.37 medium effect
**Analysis into group** *p*-value (test)	*p* = 0.0003 * (Cg) LSIL **, *p* < 0.0001(Z)	*p* = 0.14 (Cg)	*p* = 0.0015 * (Cg) LSIL **, *p* < 0.0001(Z)		
**Effect size**	phi = 2.20 large effect	phi = 1.28 large effect	phi = 2.26 large effect		
Histological diagnosis NE ^1^ Negative LSIL HSIL Adenocarcinoma	41.1% (37) 8.9% (8) 10.0% (9) 38.9% (35) 1.1% (1)	46.7% (14) 10.0% (3) 6.7% (2) 36.7% (11) 0.0% (0)	38.3% (23) 8.3% (5) 11.7% (7) 40.0% (24) 1.1% (1)	0.88 (F)	phi = 0.19 low effect
**Analysis into group** *p*-value (test)	*p* < 0.0001 * (Cg) HSIL **, *p* < 0.0001(Z)	*p* = 0.0006 * (Cg) HSIL **, *p* < 0.0001(Z)	*p* < 0.0001 * (Cg) HSIL **, *p* < 0.0001(Z)		
**Effect size**	phi = 6.60 large effect	phi = 4.56 large effect	phi = 4.94 large effect		

* = significant test (*p* < 0.05); ** = more frequent; F = Fisher’s exact test; C = chi-square goodness fit test. Z = post hoc Z-test; HSIL = high-grade squamous intraepithelial lesion, ASC-H = atypical squamous cells-cannot exclude HSIL, LSIL = low-grade squamous intraepithelial lesion, ASC-US = atypical squamous cells of undetermined significance, NE = histological diagnosis was not performed. lr = low risk; hr = high risk. 1 = the NE modality was not included in the statistical analysis performed for each column.

**Table 3 vaccines-13-00172-t003:** Cytological and histological diagnoses from the female partners of the 60 HPV-positive males in correlation with the number of genotypes identified.

Cytological Diagnosis (n = 60)	One Genotype n = 27	Two Genotypes n = 11	Three Genotypes n = 11	Four Genotypes n = 4	Five Genotypes n = 5	Six Genotypes n = 2	*p*-Value (Test)	Effect Size
Negative **^1^**	14.6% (4)	18.2% (2)	27.3% (3)	25.0% (1)	40.0% (2)	50.0% (1)	0.0496 * (F) ASC-US/four **, *p* = 0.034 ASC-US/five **, *p* = 0.0003 LSIL/two **, *p* = 0.0206 ASC-H/one **, *p* = 0.0320	phi = 3.64 large effect
ASC-US	11.1% (3)	0.0% (0)	9.1% (1)	**50.0% (2)**	**60.0% (3)**	0.0% (0)		
LSIL	25.9% (7)	**63.6% (7)**	45.5% (5)	25.0% (1)	0.0% (0)	50.0% (1)		
ASC-H	**29.6% (8)**	9.1% (1)	9.1% (1)	0.0% (0)	0.0% (0)	0.0% (0)		
HSIL	18.5% (5)	9.1% (1)	9.1% (1)	0.0% (0)	0.0% (0)	0.0% (0)		
**Analysis into group** ***p*-value (test)**	0.06 (Cg)	0.0007 * (Cg) LSIL **, *p* < 0.0001	0.21 (Cg)	n < 10 ^+^	n < 10 ^+^	n < 10 ^+^		
**Effect size**	phi = 0.61 large effect	phi = 4.22 large effect	phi = 1.75 large effect	─	─	─		
**Histological Diagnosis (n = 37)**	**One** **Genotype** **n = 20**	**Two** **Genotypes** **n = 5**	**Three** **Genotypes** **n = 7**	**Four** **Genotypes** **n = 2**	**Five** **Genotypes** **n = 2**	**Six** **Genotypes** **n = 1**	***p*-value (Test)**	**Effect size**
Negative **^1^**	10.0% (2)	0.0% (0)	14.3% (1)	50.0% (1)	0.0% (0)	100% (1)	0.33 (F)	phi = 2.68 large effect
LSIL	15.0% (3)	0.0% (0)	28.6% (2)	50.0% (1)	50.0% (1)	0.0% (0)		
HSIL	**70.0% (14)**	100% (5)	57.1% (4)	0.0% (0)	50.0% (1)	0.0% (0)		
Adenocarcinoma	5.0% (1)	0.0% (0)	0.0% (0)	0.0% (0)	0.0% (0)	0.0% (0)		
**Analysis into group** ***p*-value (test)**	<0.0001 * (Cg) HSIL **, *p* < 0.0001	n < 10 ^+^	n < 10 ^+^	n < 10 ^+^	n < 10 ^+^	n < 10 ^+^		
**Effect size**	phi = 4.92 large effect	─	─	─	─	─		

1 = the negative diagnosis was not included in the statistical analysis performed for each column. + = n < 10, test not performed. ASC-US atypical squamous cells of undetermined significance, LSIL = low-grade squamous intraepithelial lesion, HSIL= high-grade squamous intraepithelial lesion, ASC-H = atypical squamous cells-cannot exclude HSIL and adenocarcinoma. * = significant test (*p* < 0.05); ** = more frequent; C = chi-square test; F = Fisher’s exact test; Cg = chi-square goodness fit test.

**Table 4 vaccines-13-00172-t004:** Analysis of male single genotypes infection included or not in quadrivalent and/or nonavalent vaccine, concerning cytological and histological diagnosis of the female partner.

Genotype of Male Partner	Genotype Risk	Genotype Included in Vaccine	Cytological Diagnosis	Histological Diagnosis
16	High	Qv: yes; Nv: yes	ASC-H = 2 HSIL = 1	NEG = 1 HSIL = 2
31	High	Qu: no; Nv: yes	ASC-H = 1	HSIL = 1
51	High	Qv: no; Nv: no	NEG = 1 LSIL = 2 ASC-H = 1	NE = 2 LSIL = 1 HSIL = 1
52	High	Qv: no; Nv: yes	NEG = 1 LSIL = 1 ASC-H = 1	HSIL = 2 Adenocarcinoma = 1
53	High	Qv: no; Nv: no	LSIL = 2 ASC-H = 1	NE = 2 NEG = 1
56	High	Qv: no; Nv: no	ASC-H = 1 HSIL = 1	HSIL = 2
58	High	Qv: no; Nv: yes	ASC-US = 1 HSIL = 1	HSIL = 2
59	High	Qv: no; Nv: no	HSIL = 1	HSIL = 1
66	High	Qv: no; Nv: no	ASC-US = 1	LSIL = 1
68	High	Qv: no; Nv: no	ASC-H = 1	HSIL = 1
73	High	Qv: no; Nv: no	LSIL = 1	LSIL = 1
42	Low	Qv: no; Nv: no	NEG= 1	NE= 1
54	Low	Qv: no; Nv: no	ASC-US = 1	NE = 1
61	Low	Qv: no; Nv: no	LSIL = 1 HSIL = 1	HSIL = 2
83	Low	Qv: no; Nv: no	NEG = 1	NE = 1

HSIL = high-grade squamous intraepithelial lesion, ASC-H = atypical squamous cells-cannot exclude HSIL, LSIL = low-grade squamous intraepithelial lesion, ASC-US = atypical squamous cells of undetermined significance, NE = histological diagnosis was not performed. Qv = quadrivalent vaccine; Nv = nonavalent vaccine.

**Table 5 vaccines-13-00172-t005:** Analysis of vaccine and non-vaccine genotypes respect the histological lesions.

Histological Diagnosis	Male Genotypes	Quadrivalent	Nonavalent
LSIL	6-54-73	6	6
LSIL	18-33-52-66-73	18	18-33-52
LSIL	31-35-40-42	Not included	31
LSIL	31-42-53	Not included	31
LSIL	73	Not included	Not included
LSIL	51	Not included	Not included
LSIL	66	Not included	Not included
HSIL	51	Not included	Not included
HSIL	44-53	Not included	Not included
HSIL	59	Not included	Not included
HSIL	61	Not included	Not included
HSIL	42-59-61	Not included	Not included
HSIL	56	Not included	Not included
HSIL	61	Not included	Not included
HSIL	56	Not included	Not included
HSIL	68	Not included	Not included
HSIL	52-68	Not included	52
HSIL	16	16	16
HSIL	16	16	16
HSIL	58	Not included	58
HSIL	52	Not included	52
HSIL	51-31	Not included	31
HSIL	31-45	Not included	31-45
HSIL	45-68-42	Not included	45
HSIL	31	Not included	31
HSIL	52	Not included	52
HSIL	16-68	16	16
HSIL	58	Not included	58
HSIL	31-52-73	Not included	31-52
HSIL	31-42-61	Not included	31
HSIL	16-42-52-58-66	16	16-52-58
Adenocarcinoma	52	Not included	52

HSIL = high-grade squamous intraepithelial lesion, LSIL = low-grade squamous intraepithelial lesion.

**Table 6 vaccines-13-00172-t006:** Additional Absolute Impact (AAI) of the nonavalent vaccine compared to the quadrivalent vaccine.

**Histological Diagnosis: Nonavalent-Quadrivalent**	
% Absolute additional Impact (AAI)	31.1% (20 − 6 = 14/45)
**Cytological Diagnosis: Nonavalent-Quadrivalent**	
% Absolute additional Impact (AAI)	32.4% (27 − 5 = 22/68)

*p* = 0.89.

**Table 7 vaccines-13-00172-t007:** The possible role of vaccination on each cervical lesion.

	Total (n = 90)	Quadrivalent			Nonavalent		
Cytological Diagnosis		Not Included (78)	Included (12)	Not Incl. vs. Incl. *p*-Value (Test) Effect Size	Not Included (56)	Included (34)	Not Incl. vs. Incl. *p*-Value (Test) Effect Size
Negative	24.4% (22)	21.1% (19/90)	3.3% (3/90)	1.0 (F) phi = 0.0002, trivial effect	16.7% (15/90)	7.8% (7/90)	0.44 (C) phi = 0.005, trivial effect
Positive	75.6% (68)	65.6% (59/90)	10.0% (9/90)		45.6% (41/90)	30.0% (27/90)	
**Analysis** **into group** *p*-value (test)	*p* < 0.0001 * (B)	*p* < 0.0001 * (B)	*p* = 0.08 (B)		*p* = 0.0005 * (B)	*p* = 0.0008 * (B)	
Effect size	G = 0.26, large effect	G = 0.26, large effect	G = 0.25, large effect		G = 0.23, medium effect	G = 0.29, large effect	
**Histological diagnosis**	**Total** **(n = 90)**	**Not included (45)**	**Included** **(8)**		**Not included** **(31)**	**Included** **(22)**	
NE	41.1% (37)						
Negative	8.9% (8)	11.3% (6/53)	3.8% (2/53)	0.59 (F) phi = 0.10, trivial effect	11.3% (6/53)	3.8% (2/53)	0.45 (F) phi = 0.01, trivial effect
Positive	45.0% (45)	73.6% (39/53)	11.3% (6/53)		47.2% (25/53)	37.7% (20/53)	
**Analysis** **into group** *p*-value (test)	*p* < 0.0001 * (B)	*p* < 0.0001 * (B)	*p* = 0.16 (B)		*p* = 0.0007 * (B)	*p* = 0.0001 * (B)	
Effect size	G = 0.52, large effect	G = 0.37, large effect	G = 0.25, large effect		G = 0.31, large effect	G = 0.41, large effect	

NE = Not Performed, Positive = included: ASC-US atypical squamous cells of undetermined significance, LSIL = low-grade squamous intraepithelial lesion, HSIL = high-grade squamous intraepithelial lesion, ASC-H = atypical squamous cells-cannot exclude HSIL, and adenocarcinoma. * = significant test (*p* < 0.05); C = chi-square test; F = Fisher’s exact test; B = binomial test.

## Data Availability

Data are contained within the article.

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
