# Peer review of "Human Papillomavirus Infection in Partners of Women Attending Cervical Cancer Screening: A Pilot Study on Prevalence, Distribution, and Potential Use of Vaccines"

_vaccines, 2025, doi:10.3390/vaccines13020172_

Round 1

Reviewer 1 Report

Comments and Suggestions for Authors

The authors presented the article entitled “Human Papillomavirus infection in partners of women attending Cervical Cancer Screening: a pilot study on prevalence, distribution, and potential use of vaccines”.

I have suggestions.

1. In Materials and Methods section, the criteria for male cases included was not clear enough. Randomized or continued or by numerical or by other methods among the female partners with positivity hrHPV test?

2. cytologic report is a screening but histologic study is a diagnosis; clarify the roles of screening and diagnosis.

Comments on the Quality of English Language

The English could be improved to more clearly express the research.

Reviewer 2 Report

Comments and Suggestions for Authors

Dear Authors,
the paper is very interesting because it deals with a topic that has been little studied until now.
The topic is very pertinent because vaccination in men is essential to attack the human papillomavirus and prevent cancer.
I would like to make a few comments:
The target population is the one from which the inference of the results is to be made. The study population is the population to which one has access.
It is from this study population that the sample is extracted, provided that one does not have access to the entire study population.
In this paper, the study population is referred to when the sample is performed. So I wonder if an analysis of the sample size has been performed.
If so, it should be explained how this study was done and how the two groups were chosen.
The standard deviation is a measure of dispersion. It should not be put with the +- sign. In the tables it appears well in parentheses. The Standard Error is a measure of precision and can be put with the +- sign.
In health sciences, it is very difficult for a variable to have a normal distribution. The job of health professionals is precisely to ensure that this normality is not met.
In addition to normality, in order to perform parametric tests, there must be equality of variances, and the Levene test is never performed. I recommend always performing non-parametric tests. If there is normality, they will have the same results as parametric tests. If there is no normality, the non-parametric test is better.

Figure 1 is difficult to interpret the p value. Looking at the figure separately from the text, I did not understand what this p value meant.
In Table 3, care must be taken when interpreting the p values, especially in cases where there are many 0s. When a variable has several categories with 0s, it is possible to detect differences where in reality there are none, or vice versa.
I recommend taking into account that, although the correlations are statistically significant, they are small. A Rho coefficient of -0.3 is small even if it is statistically significant. In conclusion, it is said that there is a correlation. I recommend that care be taken and that this correlation is explained as small.

Reviewer 3 Report

Comments and Suggestions for Authors

In the manuscript entitled “Human Papillomavirus infection in partners of women attending Cervical Cancer Screening: a pilot study on prevalence, distribution, and potential use of vaccines” the authors present significant data concerning the distribution of HPV genotypes in men partners of HPV positive women. The authors highlight the reasons why HPV screening and HPV vaccination in male population can contribute to the elimination of viral infection. The manuscript is well written and data are sufficiently presented. However, some points need to be addressed.

The authors should provide more details concerning the natural history of HPV infection. A short description of viral DNA structure and the molecular mechanisms involved in HPV induced carcinogenesis are required to be mentioned in order to help readers to better understand the biology of HPVs.  Moreover, the authors should provide more details about the classification of HPVs into genera as well as they are required to describe how HPVs of genus Alpha are grouped into high risk and low risk genotypes. Please provide the appropriate references (Viruses. 2022 Dec 31;15(1):141. doi: 10.3390/v15010141, Viruses. 2018 Feb 13;10(2):80. doi: 10.3390/v10020080).

Lines 136-137: The reference must follow the guidelines of the journal. Please revise. 

The authors mentioned that only 5.6% (5/90) of participants (all HPV-positive) exhibited symptoms during the examination. Please provide more details about these symptoms. It would be interesting to examine which HPV genotypes are detected in these cases as well as to elucidate whether specific HPV genotypes are prevalent in this cohort.

Round 2

Reviewer 2 Report

Comments and Suggestions for Authors

Dear Authors,
Thank you very much for clarifying my comments and doubts.
The paper has improved a lot.
Congratulations